# Learning Composable Energy Surrogates
# for PDE Order Reduction

**Alex Beatson**
Princeton University
abeatson@princeton.edu

**Jordan T. Ash**
Microsoft Research NYC
ash.jordan@microsoft.com

**Geoffrey Roeder**
Princeton University
roeder@princeton.edu

**Tianju Xue**
Princeton University
txue@princeton.edu

**Ryan P. Adams**
Princeton University
rpa@princeton.edu

## Abstract

Meta-materials are an important emerging class of engineered materials in which complex macroscopic behaviour–whether electromagnetic, thermal, or mechanical–arises from modular substructure. Simulation and optimization of these materials are computationally challenging, as rich substructures necessitate high-fidelity finite element meshes to solve the governing PDEs. To address this, we leverage *parametric* modular structure to learn component-level surrogates, enabling cheaper high-fidelity simulation. We use a neural network to model the stored potential energy in a component given boundary conditions. This yields a structured prediction task: macroscopic behavior is determined by the minimizer of the system's total potential energy, which can be approximated by composing these surrogate models. Composable energy surrogates thus permit simulation in the reduced basis of component boundaries. Costly ground-truth simulation of the full structure is avoided, as training data are generated by performing finite element analysis of individual components. Using dataset aggregation to choose training data allows us to learn energy surrogates which produce accurate macroscopic behavior when composed, accelerating simulation of parametric meta-materials.

## 1 Introduction

Many physical, biological, and mathematical systems can be modeled by partial differential equations (PDEs). Analytic solutions are rarely available for PDEs of practical importance; thus, computational methods to approximate PDE solutions are critical for many problems in science and engineering. Finite element analysis (FEA) is one of the most widely used techniques for solving PDEs on spatial domains; the continuous problem is discretized and replaced by basis functions on a mesh.

The accuracy of FEA and related methods requires a sufficiently fine discrete approximation, i.e., finite element mesh. Complicated domains can require fine meshes that make it prohibitively expensive to solve the PDE. This problem is compounded for parameter identification or design optimization, where the PDE must be repeatedly solved in the inner loop of a bi-level optimization problem.

An important domain where this challenge is particularly relevant is in modeling mechanical meta-materials. Meta-materials are solids in which microstructure leads to rich spaces of macroscopic behavior, which can achieve electromagnetic and/or mechanical properties that are impossible with homogenous materials and traditional design approaches (Poddubny et al., 2013; Cai and Shalaev, 2010; Bertoldi et al., 2017). We focus on the *cellular* mechanical meta-materials proposed by Overvelde and Bertoldi (2014), which promise new high-performance materials for soft robotics and other domains (see Sec 3). Simulation of these meta-materials is challenging due to the need to

accurately capture microstructure and small-scale nonlinear elastic behavior. Finite element methods have limited ability to scale to these problems, and automated meta-material design demands accurate, efficient approximate solutions to the associated PDE.

We develop a framework which exploits spatially local structure in large-scale optimization problems—here the minimization of energy as a function of meta-material displacements. Only a small subset of material displacements are of interest, so we "collapse out" the remainder using a learned surrogate. Given a component with substructure defined by local parameters, the surrogate produces an accurate proxy energy in terms of the displacement of the component boundary. A single surrogate can be trained then used to predict energy in a larger solid by summing energies of sub-components. This allows solving the PDE in a reduced basis of component boundaries by minimizing this sum.

Other methods exist for reducing the solution cost of large PDEs. One such is the boundary element method (Aliabadi, 2002), which as with our method "collapses out" the internal degrees of freedom in a PDE leaving a problem in terms of the solution on the boundary. Unlike our method, this is performed analytically and is typically only valid for linear PDEs. Our method might be seen as a *learned* boundary element method for a particular parametric class of nonlinear PDEs. Another related line of work is homogenization. Whether micro-scale effects are modeled with fine-resolution FEM (Schröder, 2014) or a neural network (Xue et al., 2020), homogenized models require a PDE formed of homogenous representative volume elements (RVEs), and are accurate only as the ratio between the size of the RVE and the size of the macro-scale problem tends to zero.

Some approaches amortize PDE solving more directly, using neural networks to map from PDE parameters to solutions (Zhu et al., 2019; Nie et al., 2020) or constructing reduced bases via solving eigenvalue problems or interpolating between snapshots (Berkooz et al., 1993; Chatterjee, 2000). These approaches typically require solving full systems to produce training data. Our framework uses the modular decomposition of energy to train surrogate models on data generated by querying the finite element "expert" on the energy in small components, avoiding performing FEA on large systems which are expensive to solve.

## 2 Learning to optimize in collapsed bases

Solving PDEs like those that govern meta-material behavior involves finding a solution $u$ which minimizes an energy $E(u)$ subject to constraints. For mechanical meta-materials, $E(u)$ is the stored elastic potential energy in the material. We propose a framework for amortizing high-dimensional optimization problems where the objective has special conditional independence structure, such as that found in solving these PDEs. Consider the general problem of solving

$$u^* = \arg \min E(u) \,. \tag{1}$$

$u$ may be a vector in $\mathbb{R}^d$ or may belong to a richer space of functions. Often we are interested in a subset of the vector $u^*$, or the values the function $u^*$ takes on a small subdomain. To reflect this, view the solution space as the Cartesian product of a space of primary interest and a "nuisance" space. Denote the solutions as concatenations $u = [x, y]$ where $y$ is the object of interest, and $x$ is the object whose value is not of interest to an application. $x$ is roughly similar to auxiliary variables that appear in probabilistic models, but are marginalized away or discarded from the simulation. We use this decomposition to frame Eq. 1 as a bi-level optimization problem:

$$y^* = \arg \min_y \min_x E(x, y) \,. \tag{2}$$

Consider the *collapsed objective*, $\tilde{E}(y) = \min_x E(x, y)$. If $\tilde{E}(y)$ can be queried without representing $x$, we may perform *collapsed optimization* in the reduced basis of $y$, avoiding optimization in the larger basis of $u$ (Eq. 1), or performing bi-level optimization (Eq. 2). However, $\tilde{E}$ is not usually available in closed form. We consider approximating $\tilde{E}(y)$ via supervised learning. In general, this would require solving $\tilde{E} = \min_x E(x, y)$ for each example $y$ we wish to include in our training set. This is the procedure used by many surrogate-based optimization techniques (Queipo et al., 2005; Forrester and Keane, 2009; Shahriari et al., 2015). The high cost of gathering each training example makes this prohibitive when $x$ is high dimensional (and minimization is difficult) or when $y$ is high dimensional (and supervised learning requires many examples). Compositional structure in $E$ may assist us with approximating $\tilde{E}$. Many objectives may be represented as a sum:

$$E(x, y) = \sum_i E_i(x_i, y) \,. \tag{3}$$

Given this decomposition, $x_i$ and $x_j$ are conditionally independent given $y$; i.e., if we constrain $x_i$ and $y$ to take some values and perform minimization, the resulting $x_j$ or $E_j(x_j, y)$ do not vary with the value chosen for $x_i$. This follows from the partial derivative structure $\frac{\partial E_i}{\partial x_j} = 0$ for $i \neq j$.

We propose to *learn* a collapsed objective $\tilde{E}$, which exploits conditional independence structure by representing $\tilde{E}(y) = \sum_i \tilde{E}_i(y)$. This representation as a sum allows us to use $\min_{x_i} E_i(x_i, y)$ as targets for supervision, which may be found more cheaply than performing a full minimization. The learned approximations to $\tilde{E}_i$ may be composed to form an energy function with larger domain.

The language we use to describe this decomposition is chosen to reflect the conceptual similarity of our framework to *collapsed variational inference* (Teh et al., 2007) and *collapsed Gibbs sampling* (Geman and Geman, 1984; Liu, 1994), in which conditional independence allows optimization or sampling to proceed in a collapsed space where nuisance random variables are marginalized out of the relevant densities. We exploit similar structure to these techniques, albeit in a deterministic setting. Other approaches to accelerating Eq. 2 which do not exploit (3) or directly model $\tilde{E}(y)$ include amortizing the inner optimization by predicting $x^*(y) = \arg\min_x E(x, y)$ (Kingma and Welling, 2013; Brock et al., 2017), or truncation of the inner loop, either deterministic (Wu et al., 2018; Shaban et al., 2018) or randomized to reduce bias (Tallec and Ollivier, 2017; Beatson and Adams, 2019).

The optimization procedure we accelerate is the simulation of mechanical materials, where the objective corresponds to a physically meaningful energy, and the conditional independence structure arises from spatial decomposition of the domain and spatial locality of the energy density. We believe this spatial decomposition of domain and energy could be generalized to learn collapsed energies for solving many other PDEs in reduced bases. This collapsed-basis approach may also be applicable to other bi-level optimization problems where the objective decomposes as a sum of local terms.

## 3 Mechanical meta-materials

Meta-materials are engineered materials with microstructure which results in macroscopic behavior not found in nature. The most popularly known are electromagnetic meta-materials such as negative refraction index solids and "invisibility cloaks" which conceal an object through engineered distortion of electromagnetic fields (Poddubny et al., 2013; Cai and Shalaev, 2010). However, they also hold great promise in other domains: *mechanical* meta-materials use substructure to achieve unusual macroscopic behavior such as negative Poisson's ratio and nonlinear elastic responses; pores and lattices undergo reversible collapse under large deformation, enabling the engineering of complex physical affordances in soft robotics (Bertoldi et al., 2017).

Meta-materials hold promise for modern engineering design but are challenging to simulate as the microstructure necessitates a very fine finite element mesh, and as the nonlinear response makes them difficult to approximate with a macroscopic material model. Most work on meta-materials has relied on engineers and scientists to hand-design materials, rather than numerically optimizing substructure to maximize some objective (Ion et al., 2016).

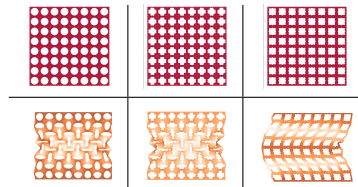

Figure 1: Cellular meta-materials. Top: at rest. Bottom: under compression, exhibiting periodic instability varying with pore shape. The left two structures exhibit negative Poisson's ratio, which does not occur in nature.

We focus on building surrogate models for the two-dimensional cellular solids investigated in Overvelde and Bertoldi (2014). These meta-materials consist of square "cells" of elastomer, each of which has a pore in its center. The pore shapes are defined by parameters $\alpha$ and $\beta$ which characterize the pore shape in polar coordinates: $r(\theta) = r_0[1 + \alpha\cos(4\theta) + \beta\cos(8\theta)]$. The parameter $r_0$ is chosen such that the pore covers half the cell's volume: $r_0 = L_0/\sqrt{\pi(2+\alpha^2+\beta^2)}$. Constraints are placed on $\alpha$ and $\beta$ to enforce a minimum material thicknesses and ensure that $\min_\theta r(\theta) > 0$ as in Overvelde and Bertoldi (2014).

These pore shapes give rise to complicated nonlinear elastic behavior, including negative Poisson's ratio and double energy wells (i.e., stored elastic energy which does not increase monotonically with strain). Realizations of this class of materials are shown under axial strain in Figure 1. The continuum mechanics behavior of these elastomer meta-materials can be captured by a neo-Hookean energy model (Ogden, 1997). Let $X \in \mathbb{R}^d$, where $d \leq 3$ in physical problems, be a point in the resting undeformed material reference configuration, and $u(X)$ be the displacement of this point

from reference configuration. The stored energy in a neo-Hookean solid is $E = \int_\Omega W(u)dX$, where $W(u)$ is a scalar energy density over $\Omega$, defined for bulk and shear moduli $\mu$ and $\kappa$ as:

$$W = \frac{\mu}{2}\left((\det F)^{-2/d}\mathrm{tr}(FF^T) - d\right) + \frac{\kappa}{2}(\det F - 1)^2 \tag{4}$$

where $F$ is the deformation gradient, $F(X) = \frac{\partial u(X)}{\partial X} + I$. Pores influence the structure of these equations by changing the material domain $\Omega$. These solids can be simulated by solving:

$$\mathrm{Div}\, S = 0 \quad X \in \Omega \tag{5}$$

$$G(u) = 0 \quad X \in \partial\Omega \tag{6}$$

where $S = \frac{\partial W}{\partial F}$ is known as the first Piola-Kirchoff stress, and where Eq. 6 defines a boundary condition. E.g. $G(u) = u - u_b$ is a Dirichlet boundary condition; in our case, an externally imposed displacement. $G(u) = \frac{\partial W}{\partial u} - f_b$ corresponds to an external force exerting a pressure on the boundary.

To simulate these meta-materials, Eq. 5 is typically solved via finite element analysis. Solving with large meta-material structures is computationally challenging due to fine mesh needed to capture pore geometry and due to the nonlinear response induced by buckling under large displacements.

Solving the PDE in Eq. 5 corresponds to finding the $u$ which minimizes the stored energy in the material subject to boundary conditions. That is, Eqs. 5 and 6 may be equivalently be expressed in an energy minimization form:

$$u = \arg\min \int_{X \in \Omega} W(u)dX \qquad \text{subject to } G(u) = 0 \in \partial\Omega \tag{7}$$

We use this form to learn surrogates which solve the PDE in a reduced basis of cell boundaries.

## 4 Composable energy surrogates

We apply the idea of learning collapsed objectives to the problem of simulating two-dimensional cellular mechanical meta-material behavior. The material response is determined by the displacement field $u$ which minimizes the energy $\int_\Omega W dX$, subject to boundary conditions. We divide $\Omega$ into regular square subregions $\Omega_i$, which we choose to be cells with $2 \times 2$ arrays of pores, and denote the intersection of the subregion boundaries with $\mathcal{B} = \partial\Omega_1 \cup \partial\Omega_2 \cup \ldots$ We let $u_i$ be the restriction of $u$ to $\Omega_i$. We take the quantity of interest to be $u_\mathcal{B}$, the restriction of $u$ to $\mathcal{B}$, and the nuisance variables to be the restriction of $u$ to $\Omega \backslash \mathcal{B}$. The partitioning of $\Omega$ is shown in Figure 2.

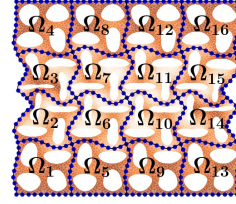

Figure 2: Meta-material domain $\Omega$, partitioned into $\Omega_1$ to $\Omega_{16}$. Black lines show $\mathcal{B}$. Blue points are control points of splines used to represent $\tilde{u}$.

The total energy decomposes as a sum over regions:

$$E(u) = \int_{X \in \Omega} W(u)dX \qquad = \sum_i \int_{X \in \Omega_i} W(u_i)dX := \sum_i E(u_i)$$

Let $\tilde{u}_i$ be the restriction of $u$ to $\partial\Omega_i$. Note $\partial\Omega_i = \mathcal{B} \cap \Omega_i$. Let the collapsed component energy be:

$$\tilde{E}_i(\tilde{u}_i) := \min_{u_i} E(u_i) \quad \text{subject to } u_i(X) = \tilde{u}_i(X) \quad X \in \partial\Omega_i.$$

This quantity is the lowest energy achievable by displacements of the *interior* of the cell $\Omega_i$, given the boundary conditions specified by $\tilde{u}_i$ on $\partial\Omega_i$. $\tilde{E}_i(\tilde{u}_i)$ depends on the shape of the region $\Omega_i$, i.e., on the geometry of the pores. Rather than each possible pore shape having a unique collapsed energy function, we introduce the pore shape parameter $\xi = (\alpha, \beta)$ as an argument, replacing $\tilde{E}_i(\tilde{u}_i)$ with $\tilde{E}(\tilde{u}_i, \xi_i)$. The macroscopic behavior of the material is fully determined by this *single* collapsed energy function $\tilde{E}(\tilde{u}_i, \xi_i)$. Given the true collapsed energy functions, we could accurately simulate material behavior in the reduced basis of the boundaries between each component $\Omega_i$.[1]

We learn to approximate this collapsed energy function from data. This function may be duplicated and composed to simulate the material in the reduced basis $\mathcal{B}$, an approach we term *composable energy surrogates* (CESs). A single CES is trained to approximate the function $\tilde{E}$ by fitting to

supervised data $(\tilde{u}_i, \xi_i, \tilde{E}(\tilde{u}_i, \xi_i))$, where $\xi_i$ and $\tilde{u}_i$ may be drawn from any distribution corresponding to anticipated pore shapes and displacements, and the targets $\tilde{E}(\tilde{u}_i, \xi_i)$ are generated by solving the PDE in a small region $\Omega_i$ with geometry defined by $\xi_i$ and with $\tilde{u}_i$ imposed as a boundary condition. This CES may be used to approximate the energy in multiple spatial locations: it may be "composed" to approximate the total energy of larger cellular meta-materials.

To efficiently solve for a reduced-basis displacement $u_{\mathcal{B}}$ on $\mathcal{B}$, we minimize the composed surrogate energy, $\hat{E}(u_{\mathcal{B}}) = \sum_i \hat{E}(\tilde{u}_i, \xi_i)$, where $\hat{E}(\tilde{u}_i, \xi_i)$ is the model's prediction of $\tilde{E}(\tilde{u}_i, \xi_i)$, the collapsed energy of one component. Training CES which produce accurate reduced-basis solutions may be thought of as a highly-structured imitation learning problem. A sufficient condition for finding the correct minimum is for the "action" taken by the surrogate—the derivative of the energy approximation $\nabla_{u_{\mathcal{B}}}\hat{E}$—to match the "action" taken by an expert—the *total* derivative, $\nabla_{u_{\mathcal{B}}} \min_{u \notin \mathcal{B}} E(u)$—along the optimization trajectory. If so, the surrogate will follow the trajectory of a valid, if non-standard, bilevel gradient-based procedure for minimizing the energy, corresponding to (2). Given an imperfect surrogate, the error in the final solution will depend on the error in approximating $\nabla_{u_{\mathcal{B}}} \min_{u \notin \mathcal{B}} E(u)$ with $\nabla_{u_{\mathcal{B}}}\hat{E}$ along the trajectory. This observation informs our model, training, and data collection procedures, described in the following sections.

## 5 Model architecture

Our CESs take the form of a neural architecture, designed to respect known properties of the true potential energy and to maximize usefulness as surrogate energy to be minimized via a gradient-based procedure. The effects of these design choices are quantified via an ablation study in the appendix.

**Reduced-basis parameterization**. We use one cubic spline for each horizontal and vertical displacement function along each face of the square, with evenly spaced control points and "not-a-knot" boundary conditions. Our vector representation of $\tilde{u}$ is $\mathbf{u} \in \mathbb{R}^{2n}$, formed from the horizontal and the vertical displacement values at each of the $n$ control points. Splines on adjacent faces share a control point at the corner. Using $N$ control points to parameterize the function along each face requires $n = 4 * (N-1)$ control points to parameterize a $1d$ function around a single cell. For all experiments we use $N = 10$ control points along each edge, resulting in $\mathbf{u} \in \mathbb{R}^{72}$.

**Model structure and loss**. Our model structure and losses are shown below. In the energy model $\hat{E}$, $f_\phi$ is a neural network with parameters $\phi$ and $\mathcal{R}$ removes rigid-body rotation and translation. Our loss function is $\mathcal{L} = \mathcal{L}^0 + \mathcal{L}^1 + \mathcal{L}^2$, which is a weighted sum of losses on the 0th, 1st and 2nd energy derivatives. $\nabla_{\mathbf{u}}$ and $\nabla_{\mathbf{u}}^2$ are the gradient and Hessian of the surrogate energy $\hat{E}$ or the ground-truth energy $\tilde{E}$ with respect to $\mathbf{u}$, and $v$ is sampled independently for each training example in a batch.

$$\hat{E}(\mathbf{u}, \xi) = \underbrace{||\mathcal{R}(\mathbf{u})||_2^2}_{\text{Linear elastic component}} \underbrace{\exp\{f_\phi(\mathcal{R}(\mathbf{u}), \xi)\}}_{\text{Stiffness}}, \qquad \mathcal{L}^0 = \underbrace{\left\|f_\phi(\mathcal{R}(\mathbf{u}), \xi) - \log \frac{\tilde{E}(\tilde{u})}{||\mathcal{R}(\mathbf{u})||_2^2}\right\|_2^2}_{\text{Log-stiffness loss}},$$

$$\mathcal{L}^1 = \underbrace{1 - \frac{\langle \nabla_{\mathbf{u}}\hat{E}, \nabla_{\mathbf{u}}\tilde{E} \rangle}{||\nabla_{\mathbf{u}}\hat{E}|| ||\nabla_{\mathbf{u}}\tilde{E}||}}_{\text{Cosine distance between gradients}}, \qquad \mathcal{L}^2 = \underbrace{1 - \frac{\langle \nabla_{\mathbf{u}}^2\hat{E}v, \nabla_{\mathbf{u}}^2\tilde{E}v \rangle}{||\nabla_{\mathbf{u}}^2\hat{E}v|| ||\nabla_{\mathbf{u}}^2\tilde{E}v||}}_{\substack{\text{Cosine distance between} \\ \text{Hessian-vector products}}} \qquad \underbrace{v \sim \mathcal{N}(0, I^{2n})}_{\text{Projection vector for Hessian}}.$$

**Invariance to rigid body transforms**. The true elastic energy is invariant to rigid body transforms of a solid. This invariance may be hard to learn exactly from data. We use a module $\mathcal{R}$ which applies *Procrustes analysis*, i.e. finds and applies the rigid body transform which minimizes the Euclidean distance to a reference (we use the rest configuration). This is differentiable and closed-form.

**Encoding a linear elastic bias**. The energy is approximated well by a linear elastic model when at rest: $\tilde{E}^i(\tilde{u}_i) \approx \mathcal{R}(\mathbf{u}_i)^T A^i \mathcal{R}(\mathbf{u}_i)$ for a stiffness matrix $A^i$ depending on $\xi_i$. We scale our net's outputs by $||\mathcal{R}(\mathbf{u}_i)||_2^2$ so that it needs only capture a "scalar stiffness" $E / ||\mathcal{R}(\mathbf{u}_i)||_2^2$ accounting for the geometry of $A^i$ given $\xi_i$ and for deviation from the linear elastic model.

**Parameterizing the log-stiffness**. The energy of a component $\tilde{E}^i(u_{0,i})$ is nonnegative, and the ratio of energy to a linear elastic approximation varies over many orders of magnitude. We parameterize the log of the scalar stiffness with our neural network $f_\phi$ rather than the stiffness.

**Log-stiffness loss**. We wish to find neural network parameters $\phi$ which lead to accurate energy predictions for many different orders of magnitude of energy and displacement. Minimizing the $\ell^2$ loss between predicted and true energies penalizes errors in predicting large energies more than proportional errors predicting small energies. Instead, we take the $\ell^2$ loss between the predicted log-stiffness $f_\phi(\mathcal{R}(\mathbf{u}), \xi)$ and the effective ground-truth log-stiffness, $\log \tilde{E}(\tilde{u})/||\mathcal{R}(\mathbf{u})||_2^2$.

**Sobolev training with gradients and Hessian-vector products**. "Sobolev training" on derivatives of a target function can aid generalization (Czarnecki et al., 2017). Accuracy of CES' derivatives is crucial, so we Sobolev train on energy gradients and Hessians. We obtain ground-truth gradients cheaply via the adjoint method (Lions, 1971). Given a solution $u_i$ to the PDE in $\Omega_i$ with boundary conditions $\tilde{u}_i$, the gradient $\nabla_{\tilde{u}_i} \tilde{E}_i(\tilde{u}_i)$ requires solving a linear system with the same cost as one Newton step of solving the PDE (Mitusch et al., 2019). The spline is a linear map $\mathcal{M}$ from $\mathbf{u}_i$ to $\tilde{u}_i$ in the finite element basis, so $\nabla_{\mathbf{u}_i} \tilde{E}_i(\tilde{u}_i) = \mathcal{M}^T \nabla_{\tilde{u}_i} \tilde{E}_i(\tilde{u}_i)$. The surrogate gradient, $\nabla_{\mathbf{u}_i} \hat{E}_\phi(\mathbf{u}_i, \xi_i)$, is computed with one backward pass. Given solution and gradient, we compute $\nabla_{\mathbf{u}}^2 \tilde{E}$ with one linear solve per entry of $\mathbf{u}$. As $\mathbf{u} \in \mathbb{R}^{72}$ and many more than 72 Newton steps are usually needed to solve the PDE, this does not dominate the cost of data collection. Computing the full Hessian of the surrogate energy, $\nabla_{\mathbf{u}_i}^2 \hat{E}_\phi(\mathbf{u}_i, \xi_i)$, would require $2n$ backward passes. Instead we train on Hessian-vector products, which require only one additional backward pass.

**Cosine distance loss for Sobolev training**. Energy gradient and Hessian values vary over many orders of magnitude, with higher energies leading to larger derivatives. We wish our model to be accurate across a range of operating conditions. Rather than placing an $\ell^2$ loss on the gradient and Hessian-vector products as in Czarnecki et al. (2017), we minimize the cosine distance between ground truth and approximate gradients and Hessians, which is naturally bounded in $[0, 1]$.

# 6  Data and training

Data collection has two phases. First, we collect training and validation datasets using Hamiltonian Monte Carlo (Duane et al., 1987) to preferentially sample displacements which correspond to lower energy modes. Next, we perform dataset aggregation (Ross et al., 2011) to augment the dataset so that the surrogate will be accurate on states encountered when deployed. We provide details of the hardware and the software packages used in the appendix.

**Solving the PDE**. To collect training data, we use the reduced-basis displacement $\tilde{u}$ corresponding to a vector of spline coefficients $\mathbf{u}$ as the boundary condition around a domain $\Omega$ representing a $2 \times 2$-pore subdomain, and solve the PDE using a load-stepped relaxed Newton's method (Sheng et al., 2002). The relaxed Newton's method takes the iteration $\vec{u} \leftarrow \vec{u} - \lambda (\frac{\partial^2 E}{\partial \vec{u}^2})^{-1} \frac{\partial E}{\partial \vec{u}}$. Here, $0 < \lambda < 1$ is the relaxation parameter (analogous to a step size), and $\vec{u}$ is the vector of coefficients defining $u$ in the FEA basis. Newton's method requires an initial guess which is sufficiently close to the true solution (Kythe et al., 2004). Smaller relaxation parameters yield a greater radius of convergence but necessitate more steps to solve the PDE.

The radius of convergence can also be aided by load-stepping: solving the PDE for a sequence of boundary conditions, annealing from an initial boundary condition for which we have a good initial guess (e.g., the rest configuration) to a final boundary condition $\tilde{u}$, using the solution to the previous problem as an initial guess for Newton's method for the next problem. We find that combining load stepping with a relaxed Newton's method is more efficient than using either alone. Except where specified, we linearly anneal from rest to $\tilde{u}$ over 10 load steps and use a relaxation parameter $\lambda = 0.1$.

**Initial dataset collection**. We wish to train on varied displacement boundary conditions. As solution procedures minimize energy, lower energy modes will be encountered in the solve. We choose a distribution with density the product of a Boltzmann density $\exp\{\tilde{E}\}/Z$ and a Gaussian density $\mathcal{N}(\bar{x}(\mathbf{u}); \bar{\mu}, \Sigma)$, where $\bar{x}(\mathbf{u}) \in \mathbb{R}^{2 \times 2}$ is a macroscopic strain tensor[2] corresponding to $\mathbf{u}$, $\bar{\mu}$ is a target strain drawn from an i.i.d. Gaussian with standard deviation 0.15, and $\Sigma$ is set to $(\bar{\mu} \circ \bar{\mu})^{-1}$.

Given a solution to the PDE, the log-density and its displacement may be cheaply computed (the latter via the adjoint method). Making use of these gradients, we sample data points with Hamiltonian Monte Carlo (HMC). After sampling a data point, we compute the corresponding Hessian and save the tuple $(\mathbf{u}, \xi, \tilde{E}, \nabla_{\mathbf{u}} \tilde{E}, \nabla_{\mathbf{u}}^2 \tilde{E})$ as a data point.

We initialize each HMC data collector by sampling a macroscopic displacement target and a random pore shape. We do not use load-stepping, instead using the solution for the $\mathbf{u}$ used in the previous iteration of HMC's leapfrog integration as an initial guess for solving the PDE. We randomize HMC hyperparameters for each collector to attempt to minimize the impact of specific settings: see the appendix for exact ranges. We sample 55000 training examples and 5000 validation examples altogether. We visualize displacements drawn from this distribution in the appendix.

**Data aggregation**. Surrogate deployment defies standard i.i.d. assumptions in supervised learning. The deployed surrogate encounters states determined by the energy it defines and by boundary conditions on the composed body. Given a dataset such as that we sampled with HMC, the distribution over states encountered by the surrogate in deployment may be very different to the distribution of states in this dataset.

This problem—that training an agent to predict expert actions can lead to trajectories dissimilar to those on which it was trained—is a central concern in the imitation learning literature. A number of solutions exist (Schroecker and Isbell, 2017). One is dataset aggregation, or DAGGER (Ross et al., 2011), which reduces imitation learning or structured prediction to online learning.

In DAGGER, a policy is deployed and trajectories are collected. The expert is queried on the states in these trajectories. The state-action pairs are appended to the dataset, and the policy is retrained on this dataset. This process of deployment, querying, appending data, and retraining, is iterated. The distribution of states encountered in deployment and the distribution of states in the dataset converge. Under appropriate assumptions, the instantaneous regret of the learned policy vanishes with the number of iterations, i.e., the learned policy matches the expert policy on its own trajectories.

Ross et al. (2011) present DAGGER as a method for discrete action spaces. We have a continuous action space: the gradient of the energy in a cell. We do not investigate generalizing DAGGER's regret guarantees to continuous action spaces, but the intuition holds that we wish our model to "imitate" the finite element "expert" on the optimization trajectories the model produces.

We initialize our training data with HMC as described earlier. We then apply DAGGER by iterating: (i) training the surrogate; (ii) composing surrogates and finding displacements which minimize the composed energy; (iii) sampling displacements along the surrogate's solution path, querying the ground-truth energy and energy derivatives using FEA, and adding these new data points to the dataset. We visualize displacements generated by DAGGER in the appendix.

# 7 Software and hardware

We implement the finite element models in `dolfin` (Logg and Wells, 2010; Logg et al., 2012b), a Python front end to FEniCS (Alnæs et al., 2015; Logg et al., 2012a). To differentiate through finite element solutions, we use the package `dolfin-adjoint` (Mitusch et al., 2019). We implement surrogate models in PyTorch (Paszke et al., 2019).

We use Ray (Moritz et al., 2018) to run distributed workloads on Amazon EC2. The initial dataset is collected using 80 M4.xlarge CPU spot workers. While training the surrogate, we use a GPU P3.large driver node to train the model, and 80 M4.xlarge CPU spot worker nodes performing DAGGER in parallel. These workers receive updated surrogate model parameters, compose and deploy the surrogate, sample displacements along the solution path, query the finite element model for energy and derivatives, and return data to the driver node. Initial dataset collection and model training with DAGGER each take about one day in wall-clock time.

# 8 Empirical evaluation

We demonstrate the ability of Composable Energy Surrogates (CES) to efficiently produce accurate solutions. We consider the systems constructed in Overvelde and Bertoldi (2014): structures with an $8 \times 8$ array of pores, corresponding to a $4 \times 4$ assembly of our surrogates, each representing a $2 \times 2$-pore component. We sample pore shapes from a uniform distribution over valid shapes defined in Overvelde and Bertoldi (2014). For DAGGER, we sample vertical axial strain magnitudes from $\mathcal{U}(0., 0.3)$, and apply compression with probability $0.8$ (as compressive displacements involve more interesting pore collapse) or tension with probability $0.2$.

We compare our composed surrogates to finite element analysis with different-fidelity meshes under axial compression and tension with a macroscopic displacement of $0.125L_0$, where $L_0$ is the original length of the solid. See the appendix for details of the finite element meshes. We use seven pore shapes: $\xi = (0,0)$, corresponding to circular pores, and six $\xi$ sampled from a uniform distribution over pore parameters defined as valid in Overvelde and Bertoldi (2014).

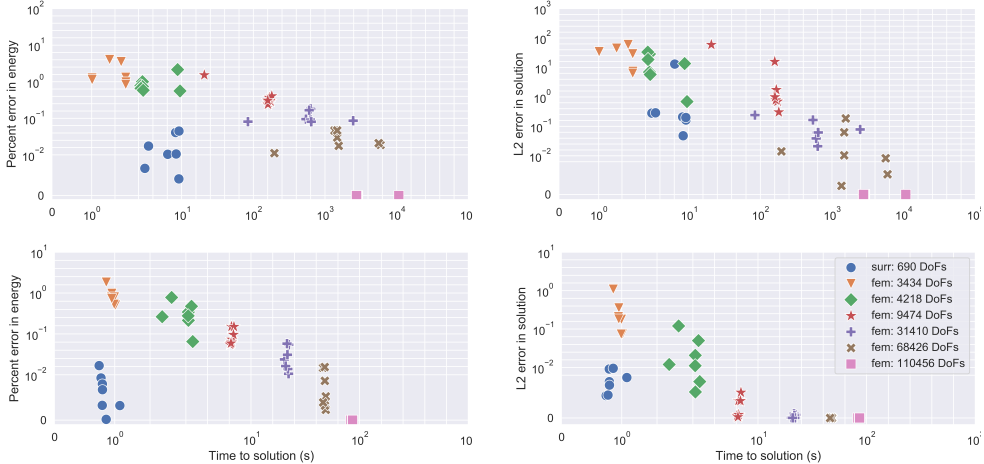

Figure 3: Error in solution and in estimated energy vs solution wall clock time for the composed energy surrogate and for finite element models with varying mesh sizes. Top: axial compression. Bottom: axial tension.

We use PyTorch's L-BFGS routine to minimize the composed surrogate energy, with step size 0.25 and default criteria for checking convergence. We attempt to solve each finite element model with FEniCS' Newton method with $[1, 2, 5, 10, 20]$ load steps and relaxation parameters $[0.9, 0.7, 0.4, 0.1, 0.05]$, and record time taken for the *fastest* convergent solve. Under compression these solids exhibit nonlinear behavior, and only more conservative solves converge. Under tension they behave closer to a linear elastic model, and Newton's method converges quickly. Measurements are taken on an AWS M4.xlarge EC2 CPU instance. Using a GPU could provide further acceleration.

We measure error in the solution and in the macroscopic energy. The former is $||\hat{u} - u^*||_2^2$, where $\hat{u}$ and $u^*$ are the approximation and ground-truth evaluated at spline control points. The latter is the relative error $|\hat{E}(\hat{u}) - E^*(u^*)|/E^*(u^*)$, where $\hat{E}(\hat{u})$ is the approximated energy of the approximate solution, and $E^*(u^*)$ is the ground-truth energy of the ground-truth solution. As the energy function determines behavior, accuracy of energy is a potential indicator of ability to generalize to larger structures. The highest-fidelity finite element model is taken as ground truth, and thus has an error of zero on both metrics. Multiple minimizers exist as energy is preserved under rigid body transforms, so before comparing a solution $\hat{u}$ to the ground-truth $u^*$ we check each vertical and horizontal flip and use the flip which minimizes the solution error.

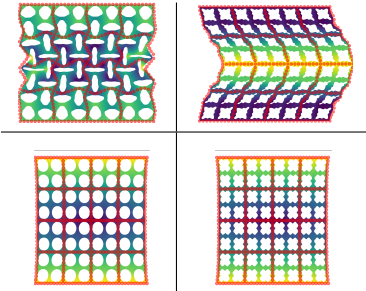

Figure 4: Meta-materials under compression (top) and tension (bottom), with solution found via CES shown in red at spline control points.

Figure 3 shows our evaluation. Composed energy surrogates are more efficient than high-fidelity FEA simulations yet more accurate than low-fidelity simulations. CES produces solutions with equivalent $\ell^2$ error to FEA solutions which use an order of magnitude more variables or computation time, and with an order of magnitude less $\ell^2$ error than FEM solutions requiring the same computation. This gap increases to several orders of magnitude when we consider percentage error in the predicted strain energy. We visualize the ground-truth and the CES approximation in Figure 4. See the appendix for visualization of FEM and CES solutions for the remaining structures.

# 9 Limitations and opportunities

**Use of DAGGER**. We use DAGGER to help CES match the ground-truth on the states encountered during the solution trajectory. This requires one to specify in advance the conditions under which the surrogate will be deployed. Investigating CES' ability to generalize to novel deployment conditions–and designing surrogates which can do so effectively–is an important direction for future work.

**Error estimation, refinement, and guarantees**. Finite element methods permit a straightforward way to estimate the error (compare to the solution in a more-refined basis) and control it (via refinement). CES currently lacks these properties.

**Finite element baseline**. There is an immense body of work on finite element methods and iterative solvers. We provide a representative baseline, but our work should not be taken as a comparison with the "state-of-the-art". We show that composable machine-learned energy surrogates enjoy advantages over a reasonable baseline, and hold promise for scalable amortization of solving modular PDEs.

**Hyperparameters**. Both our method and the finite element baseline rely on a multitude of hyperparameters: the size of the spline reduced basis; the size and learning rate of the neural network; the size and degree of the finite element approximation; and the specific variant of Newton's method to solve the finite element model. We do not attempt a formal, exhaustive search over these parameters.

**Known structure**. We leave much fruit on the vine in terms of engineering structure known from the into our surrogate. For example, one could also use a more expressive normalizer than $||\mathbf{u}||_2^2$, e.g. the energy predicted by a coarse-grained linear elastic model, or exploit spatially local correlation, e.g. by using a 1-d convolutional network around the boundary of the cell.

# 10 Conclusion

We present a framework for collapsing optimization problems with local bilevel structure by learning composable energy surrogates. This framework is applied to amortizing the solution of PDEs corresponding to mechanical meta-material behavior. Learned composable energy surrogates are more efficient than high-fidelity FEA yet more accurate than low-fidelity FEA, occupying a new point on the Pareto frontier. We believe that these surrogates could accelerate metamaterial design, as well as design and identification of other systems described by PDEs with parametric modular structure.

# 11 Acknowledgements

We would like to thank Alexander Niewiarowski for numerous helpful discussions about continuum mechanics and FEA, Ari Seff for help finding a particularly difficult bug, and Maurizio Chiaramonte for inspiring early conversations about metamaterials and model order reduction.

# 12 Financial disclosures

**Funding:** This work was funded by NSF IIS-1421780 and by the Princeton Catalysis Initiative.
**Other financial interests:** RPA is on the board of directors of Cambridge Machines Ltd., and is a scientific advisor to Manifold Bio.

# 13 Broader impacts

Our work accelerates the simulation of mechanical meta-materials, and could lead to methods for accelerated simulation of other PDEs. More efficient materials design could have impact on a wide variety of downstream applications, such as soft robotics, structural engineering, biomedical engineering, and many more. Due to the incredibly wide variety of applications which might make use of advances in material design–every physical man-made object makes use of this science–it is difficult to precisely assess impact. However, we believe that meta-material driven advances in soft robotics and structural/biomedical engineering are likely to have a range of positive effects.

## Footnotes

[1]So long as forces and constraints are only applied on $\mathcal{B}$.

[2]See the appendix for approximating $\bar{x}$ from $\mathbf{u}$.

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
