[Supplementary Material · appendix.pdf]

# Appendix

**Abstract:** This appendix consists of: (i) specification of the data generating distribution and hyper-parameters; (ii) visualization of data generated via Hamiltonian Monte Carlo; (iii) visualization of data generated via DAGGER; (iv) hyperparameters used for neural network specification and training; (v) ablation study of neural network design choices; (vi) specification of the finite element meshes used as baselines; (vii) visualization of all solutions found under compression and tension for each pore shape for each baseline mesh and the composed energy surrogate.

## 1. Data generation with Hamiltonian Monte Carlo

We use 100 data collectors, each with randomly drawn hyperparameters, which each terminate (and have their place taken by a newly initialized collector) after sampling 25 data points. We collect 60,000 data points, consisting of a training set of 55,000 and a validation set of 5,000. As our distribution is arbitrary, and as we assume that more data is always a good thing, when a HMC sample is rejected, we still add it to the dataset, but return to the last un-rejected sample to continue the Markov chain.

Hyperparameter distributions are chosen heuristically such that the finite element simulation tends to converge in a reasonable amount of time. The hyperparameter distributions are as follows:

- Leapfrog step size: $\mathcal{U}(0.005, 0.02)$
- Leapfrog path length: $\mathcal{U}(0.05, 0.3)$
- Temperature used to scale the log-probability: $\mathcal{U}([0.0001, 0.0005, 0.001, 0.005, 0.01, 0.05, 0.1])$
- Standard deviation of the Gaussian from which the Hamiltonian momentum is drawn: $\mathcal{U}(0.01, 0.3)$

We approximate the macroscopic strain tensor $\bar{x}$ from $\mathbf{u}$ as:

$$\bar{x}(\mathbf{u}) = \frac{1}{N} \begin{bmatrix} \sum_{X \in \text{rhs}} u_1(X) - \sum_{X \in \text{lhs}} u_1(x) & \sum_{X \in \text{rhs}} u_2(X) - \sum_{X \in \text{lhs}} u_2(X) \\ \sum_{X \in \text{top}} u_1(X) - \sum_{X \in \text{bot}} u_1(x) & \sum_{X \in \text{top}} u_2(X) - \sum_{X \in \text{bot}} u_2(X) \end{bmatrix}$$

Above, $u_1(X)$ and $u_2(X)$ are horizontal and vertical displacements defined by $\mathbf{u}$ at a point $X$, and top, bot, lhs and rhs are the set of control point locations for the splines on the top, bottom, left and right of the component.

## 2. Visualizing HMC data

Here we display 24 randomly chosen examples from the training set.

## 3. Visualizing DAgger data

Here we display 24 randomly chosen examples from the data collected with DAGGER.

## 4. Neural network hyperparameters

We use a fully-connected neural network with three hidden layers of 512 units, Swish nonlinarities, and He initialization. We optimize our neural network using Adam with a learning rate of 3e-4 and a batch size of 512.

## 5. Surrogate design ablation study

We perform an ablation study by switching on and off the following independent variables:

- "Scale": parameterizing the log of the scalar stiffness, vs parameterizing energy directly;
- "Remove rigid": removing rigid body translations from the data via Procrustes analysis;
- "Sobolev-G": Sobolev training on energy gradients;
- "Sobolev-Hvp": Sobolev training on energy Hessian-vector products.

We measure performance after 10,000 training steps (93 epochs) on the training set, without DAgger. We evaluate each model on the validation dataset using the following metrics:

- "E %err": the error in predicted energy, expressed as a percentage of the true energy;
- "G-sim": the cosine similarity between predicted and true gradients;
- "Hvp-sim": the cosine similarity between predicted and true Hessian-vector products.

Results are shown below. For the independent variables, a value of '1' indicates that method or technique was turned on, while a value of '0' indicates it was turned off.

Each design choice improves the validation metrics. "Remove rigid" has marginal impact, as our training displacements contain little rigid body transformation. We leave this feature in as it causes no harm; as it improved performance under earlier dataset creation methods which resulted in more rigid body translation; and as removing translations before computing energy is necessary to be able to compose energy surrogates by tiling.

| Scale | Remove rigid | Sobolev-G | Sobolev-Hvp | E %err | G-sim | Hvp-sim |
|---|---|---|---|---|---|---|
| 0 | 0 | 0 | 0 | 0.16 | 0.61 | 0.49 |
| 0 | 0 | 0 | 1 | 0.82 | 0.37 | 0.83 |
| 0 | 0 | 1 | 0 | 0.24 | 0.87 | 0.75 |
| 0 | 0 | 1 | 1 | 0.39 | 0.91 | 0.82 |
| 0 | 1 | 0 | 0 | 0.17 | 0.62 | 0.49 |
| 0 | 1 | 0 | 1 | 0.65 | 0.61 | 0.83 |
| 0 | 1 | 1 | 0 | 0.33 | 0.87 | 0.73 |
| 0 | 1 | 1 | 1 | 0.46 | 0.91 | 0.82 |
| 1 | 0 | 0 | 0 | 0.15 | 0.61 | 0.48 |
| 1 | 0 | 0 | 1 | 0.1 | 0.85 | 0.8 |
| 1 | 0 | 1 | 0 | 0.087 | 0.85 | 0.72 |
| 1 | 0 | 1 | 1 | 0.091 | 0.89 | 0.81 |
| 1 | 1 | 0 | 0 | 0.14 | 0.62 | 0.49 |
| 1 | 1 | 0 | 1 | 0.097 | 0.85 | 0.8 |
| 1 | 1 | 1 | 0 | 0.087 | 0.85 | 0.71 |
| 1 | 1 | 1 | 1 | 0.089 | 0.9 | 0.81 |

## 6. Finite element baselines

We generate meshes for the finite element baselines using two parameters: pore resolution, and minimum mesh resolution. Firstly, for each pore in the cellular solid, we generate a polygon representing that pore using a number of points equal to pore resolution. We let the material domain be the overall volume of the solid with these polygons subtracted. Next, we generate a mesh over the material domain using MSHR's automated mesh generation routine, passing as resolution minimum mesh resolution multiplied by the number of cells. In MSHR, the resolution parameter controls the maximum cell size, which is the diameter of the domain's

bounding circle divided by the resolution. It should be noted that cells can be much smaller than this maximum size, or there can be many more cells than the resolution parameter would imply, as MSHR will place one cell vertex on each point used to construct the domain geometry (i.e. each point in the pore polygon).

## 7. Benchmark visualizations

In the following pages we visualize the solutions found for each pore by each FEA mesh and by CES. For each pore we use six different finite element meshes. These respectively used $[4, 8, 16, 32, 48, 64]$ points used to define the geometry of each pore, and minimum of $[1, 2, 4, 8, 12, 16]$ internal mesh vertices along a given axis per pore. Given these parameters and the geometry of the material domain, meshes were created using the automatic mesh generation tool from mshr (the mesh generation component of FEniCS). We include the number of degrees of freedom in the finite element basis in each plot. We superimpose the solution found with CES in red dots on the solution found with FEA. The CES solution has 690 degrees of freedom in all cases.

## 7.1. Compression

Min cell res: 1; pore res: 4; DoFs: 3434; shape: (0.0048, -0.0655); FEM energy: 9.26e-02; CES energy: 3.93e-02

Min cell res: 2; pore res: 8; DoFs: 4610; shape: (0.0048, -0.0655); FEM energy: 7.98e-02; CES energy: 3.93e-02

Min cell res: 4; pore res: 16; DoFs: 10452; shape: (0.0048, -0.0655); FEM energy: 5.13e-02; CES energy: 3.93e-02

Min cell res: 8; pore res: 32; DoFs: 30290; shape: (0.0048, -0.0655); FEM energy: 4.32e-02; CES energy: 3.93e-02

Min cell res: 12; pore res: 48; DoFs: 66594; shape: (0.0048, -0.0655); FEM energy: 4.10e-02; CES energy: 3.93e-02

Min cell res: 16; pore res: 64; DoFs: 111760; shape: (0.0048, -0.0655); FEM energy: 3.98e-02; CES energy: 3.93e-02

Min cell res: 1; pore res: 4; DoFs: 3074; shape: (-0.0576, -0.0379);
FEM energy: 9.98e-02; CES energy: 4.23e-02

Min cell res: 2; pore res: 8; DoFs: 4218; shape: (-0.0576, -0.0379);
FEM energy: 7.84e-02; CES energy: 4.23e-02

Min cell res: 4; pore res: 16; DoFs: 9858; shape: (-0.0576, -0.0379);
FEM energy: 5.62e-02; CES energy: 4.23e-02

Min cell res: 8; pore res: 32; DoFs: 27724; shape: (-0.0576, -0.0379);
FEM energy: 4.83e-02; CES energy: 4.23e-02

Min cell res: 12; pore res: 48; DoFs: 67914; shape: (-0.0576, -0.0379);
FEM energy: 4.49e-02; CES energy: 4.23e-02

Min cell res: 16; pore res: 64; DoFs: 113146; shape: (-0.0576, -0.0379)
FEM energy: 4.41e-02; CES energy: 4.23e-02

Min cell res: 1; pore res: 4; DoFs: 3434; shape: (0.0242, -0.0153); FEM energy: 8.00e-02; CES energy: 4.06e-02

Min cell res: 2; pore res: 8; DoFs: 4610; shape: (0.0242, -0.0153); FEM energy: 7.01e-02; CES energy: 4.06e-02

Min cell res: 4; pore res: 16; DoFs: 9744; shape: (0.0242, -0.0153); FEM energy: 5.52e-02; CES energy: 4.06e-02

Min cell res: 8; pore res: 32; DoFs: 31906; shape: (0.0242, -0.0153); FEM energy: 4.40e-02; CES energy: 4.06e-02

Min cell res: 12; pore res: 48; DoFs: 66890; shape: (0.0242, -0.0153); FEM energy: 4.11e-02; CES energy: 4.06e-02

Min cell res: 16; pore res: 64; DoFs: 111766; shape: (0.0242, -0.0153); FEM energy: 4.03e-02; CES energy: 4.06e-02

Min cell res: 1; pore res: 4; DoFs: 3074; shape: (-0.207, 0.121);
FEM energy: 1.00e-01; CES energy: 2.18e-02

Min cell res: 2; pore res: 8; DoFs: 4602; shape: (-0.207, 0.121);
FEM energy: 3.45e-02; CES energy: 2.18e-02

Min cell res: 4; pore res: 16; DoFs: 9474; shape: (-0.207, 0.121);
FEM energy: 5.58e-02; CES energy: 2.18e-02

Min cell res: 8; pore res: 32; DoFs: 32400; shape: (-0.207, 0.121);
FEM energy: 2.61e-02; CES energy: 2.18e-02

Min cell res: 12; pore res: 48; DoFs: 64426; shape: (-0.207, 0.121);
FEM energy: 2.31e-02; CES energy: 2.18e-02

Min cell res: 16; pore res: 64; DoFs: 114460; shape: (-0.207, 0.121);
FEM energy: 2.21e-02; CES energy: 2.18e-02

Min cell res: 1; pore res: 4; DoFs: 3074; shape: (-0.0614, -0.0228);
FEM energy: 9.77e-02; CES energy: 4.36e-02

Min cell res: 2; pore res: 8; DoFs: 4218; shape: (-0.0614, -0.0228);
FEM energy: 7.59e-02; CES energy: 4.36e-02

Min cell res: 4; pore res: 16; DoFs: 9474; shape: (-0.0614, -0.0228);
FEM energy: 5.67e-02; CES energy: 4.36e-02

Min cell res: 8; pore res: 32; DoFs: 29164; shape: (-0.0614, -0.0228);
FEM energy: 4.94e-02; CES energy: 4.36e-02

Min cell res: 12; pore res: 48; DoFs: 64634; shape: (-0.0614, -0.0228);
FEM energy: 4.67e-02; CES energy: 4.36e-02

Min cell res: 16; pore res: 64; DoFs: 113802; shape: (-0.0614, -0.0228)
FEM energy: 4.57e-02; CES energy: 4.36e-02

Min cell res: 1; pore res: 4; DoFs: 2298; shape: (-0.184, -0.106);
FEM energy: 1.43e-01; CES energy: 2.86e-02

Min cell res: 2; pore res: 8; DoFs: 4218; shape: (-0.184, -0.106);
FEM energy: 8.97e-02; CES energy: 2.86e-02

Min cell res: 4; pore res: 16; DoFs: 10754; shape: (-0.184, -0.106);
FEM energy: 4.00e-02; CES energy: 2.86e-02

Min cell res: 8; pore res: 32; DoFs: 31074; shape: (-0.184, -0.106);
FEM energy: 3.32e-02; CES energy: 2.86e-02

Min cell res: 12; pore res: 48; DoFs: 66462; shape: (-0.184, -0.106);
FEM energy: 2.98e-02; CES energy: 2.86e-02

Min cell res: 16; pore res: 64; DoFs: 113008; shape: (-0.184, -0.106);
FEM energy: 2.84e-02; CES energy: 2.86e-02

Min cell res: 1; pore res: 4; DoFs: 3434; shape: (0.0, 0.0);
FEM energy: 8.16e-02; CES energy: 4.34e-02

Min cell res: 2; pore res: 8; DoFs: 4218; shape: (0.0, 0.0);
FEM energy: 7.01e-02; CES energy: 4.34e-02

Min cell res: 4; pore res: 16; DoFs: 9474; shape: (0.0, 0.0);
FEM energy: 5.78e-02; CES energy: 4.34e-02

Min cell res: 8; pore res: 32; DoFs: 31410; shape: (0.0, 0.0);
FEM energy: 4.77e-02; CES energy: 4.34e-02

Min cell res: 12; pore res: 48; DoFs: 68426; shape: (0.0, 0.0);
FEM energy: 4.47e-02; CES energy: 4.34e-02

Min cell res: 16; pore res: 64; DoFs: 110372; shape: (0.0, 0.0);
FEM energy: 4.42e-02; CES energy: 4.34e-02

## 7.2. Tension

Min cell res: 1; pore res: 4; DoFs: 3074; shape: (-0.207, 0.121);
FEM energy: 8.30e-02; CES energy: 4.12e-02

Min cell res: 2; pore res: 8; DoFs: 4602; shape: (-0.207, 0.121);
FEM energy: 4.47e-02; CES energy: 4.12e-02

Min cell res: 4; pore res: 16; DoFs: 9474; shape: (-0.207, 0.121);
FEM energy: 4.83e-02; CES energy: 4.12e-02

Min cell res: 8; pore res: 32; DoFs: 32400; shape: (-0.207, 0.121);
FEM energy: 4.42e-02; CES energy: 4.12e-02

Min cell res: 12; pore res: 48; DoFs: 64442; shape: (-0.207, 0.121);
FEM energy: 4.26e-02; CES energy: 4.12e-02

Min cell res: 16; pore res: 64; DoFs: 114482; shape: (-0.207, 0.121);
FEM energy: 4.19e-02; CES energy: 4.12e-02

Min cell res: 1; pore res: 4; DoFs: 3074; shape: (-0.0576, -0.0379);
FEM energy: 8.27e-02; CES energy: 4.63e-02

Min cell res: 2; pore res: 8; DoFs: 4218; shape: (-0.0576, -0.0379);
FEM energy: 6.20e-02; CES energy: 4.63e-02

Min cell res: 4; pore res: 16; DoFs: 9858; shape: (-0.0576, -0.0379);
FEM energy: 4.95e-02; CES energy: 4.63e-02

Min cell res: 8; pore res: 32; DoFs: 27730; shape: (-0.0576, -0.0379);
FEM energy: 4.72e-02; CES energy: 4.63e-02

Min cell res: 12; pore res: 48; DoFs: 67914; shape: (-0.0576, -0.0379);
FEM energy: 4.64e-02; CES energy: 4.63e-02

Min cell res: 16; pore res: 64; DoFs: 113146; shape: (-0.0576, -0.0379)
FEM energy: 4.61e-02; CES energy: 4.63e-02

Min cell res: 1; pore res: 4; DoFs: 2298; shape: (-0.184, -0.106);
FEM energy: 1.15e-01; CES energy: 4.09e-02

Min cell res: 2; pore res: 8; DoFs: 4218; shape: (-0.184, -0.106);
FEM energy: 7.22e-02; CES energy: 4.09e-02

Min cell res: 4; pore res: 16; DoFs: 10754; shape: (-0.184, -0.106);
FEM energy: 4.66e-02; CES energy: 4.09e-02

Min cell res: 8; pore res: 32; DoFs: 31074; shape: (-0.184, -0.106);
FEM energy: 4.29e-02; CES energy: 4.09e-02

Min cell res: 12; pore res: 48; DoFs: 66560; shape: (-0.184, -0.106);
FEM energy: 4.12e-02; CES energy: 4.09e-02

Min cell res: 16; pore res: 64; DoFs: 113048; shape: (-0.184, -0.106);
FEM energy: 4.05e-02; CES energy: 4.09e-02

Min cell res: 1; pore res: 4; DoFs: 3434; shape: (0.0048, -0.0655);
FEM energy: 7.72e-02; CES energy: 4.32e-02

Min cell res: 2; pore res: 8; DoFs: 4610; shape: (0.0048, -0.0655);
FEM energy: 6.33e-02; CES energy: 4.32e-02

Min cell res: 4; pore res: 16; DoFs: 10452; shape: (0.0048, -0.0655);
FEM energy: 4.70e-02; CES energy: 4.32e-02

Min cell res: 8; pore res: 32; DoFs: 30290; shape: (0.0048, -0.0655);
FEM energy: 4.42e-02; CES energy: 4.32e-02

Min cell res: 12; pore res: 48; DoFs: 66602; shape: (0.0048, -0.0655);
FEM energy: 4.33e-02; CES energy: 4.32e-02

Min cell res: 16; pore res: 64; DoFs: 111760; shape: (0.0048, -0.0655);
FEM energy: 4.29e-02; CES energy: 4.32e-02

Min cell res: 1; pore res: 4; DoFs: 3434; shape: (0.0, 0.0);
FEM energy: 6.86e-02; CES energy: 4.56e-02

Min cell res: 2; pore res: 8; DoFs: 4218; shape: (0.0, 0.0);
FEM energy: 5.52e-02; CES energy: 4.56e-02

Min cell res: 4; pore res: 16; DoFs: 9474; shape: (0.0, 0.0);
FEM energy: 4.80e-02; CES energy: 4.56e-02

Min cell res: 8; pore res: 32; DoFs: 31418; shape: (0.0, 0.0);
FEM energy: 4.59e-02; CES energy: 4.56e-02

Min cell res: 12; pore res: 48; DoFs: 68490; shape: (0.0, 0.0);
FEM energy: 4.56e-02; CES energy: 4.56e-02

Min cell res: 16; pore res: 64; DoFs: 110372; shape: (0.0, 0.0);
FEM energy: 4.55e-02; CES energy: 4.56e-02

Min cell res: 1; pore res: 4; DoFs: 3434; shape: (0.0242, -0.0153);
FEM energy: 6.74e-02; CES energy: 4.43e-02

Min cell res: 2; pore res: 8; DoFs: 4610; shape: (0.0242, -0.0153);
FEM energy: 5.56e-02; CES energy: 4.43e-02

Min cell res: 4; pore res: 16; DoFs: 9744; shape: (0.0242, -0.0153);
FEM energy: 4.68e-02; CES energy: 4.43e-02

Min cell res: 8; pore res: 32; DoFs: 31906; shape: (0.0242, -0.0153);
FEM energy: 4.46e-02; CES energy: 4.43e-02

Min cell res: 12; pore res: 48; DoFs: 66890; shape: (0.0242, -0.0153);
FEM energy: 4.41e-02; CES energy: 4.43e-02

Min cell res: 16; pore res: 64; DoFs: 111758; shape: (0.0242, -0.0153);
FEM energy: 4.40e-02; CES energy: 4.43e-02

Min cell res: 1; pore res: 4; DoFs: 3074; shape: (-0.0614, -0.0228);
FEM energy: 8.11e-02; CES energy: 4.69e-02

Min cell res: 2; pore res: 8; DoFs: 4218; shape: (-0.0614, -0.0228);
FEM energy: 6.00e-02; CES energy: 4.69e-02

Min cell res: 4; pore res: 16; DoFs: 9474; shape: (-0.0614, -0.0228);
FEM energy: 4.97e-02; CES energy: 4.69e-02

Min cell res: 8; pore res: 32; DoFs: 29164; shape: (-0.0614, -0.0228);
FEM energy: 4.77e-02; CES energy: 4.69e-02

Min cell res: 12; pore res: 48; DoFs: 64634; shape: (-0.0614, -0.0228);
FEM energy: 4.71e-02; CES energy: 4.69e-02

Min cell res: 16; pore res: 64; DoFs: 113818; shape: (-0.0614, -0.0228)
FEM energy: 4.69e-02; CES energy: 4.69e-02