[Reviews · NeurIPS 2020]

Review 1

Summary and Contributions: The authors describe a NN for predicting potential energy functions for macro-elements in a finite element framework in order to simulate nonlinear mechanical meta-materials. The models are parameterized by the shape of the pores in the material. The authors compare their method on a test problem against finite element models with varying levels of refinement, reporting both solution time and energy error.

Strengths: This was one of my favorite papers this year. The authors are clearly well grounded in both computational mechanics and machine learning, and I consider the work interesting from both perspectives. Other work in this space largely involves analytic homogenization theory (which is often difficult in the face of microscopic nonlinear effects like buckling) or computational homogenization techniques like two-scale finite elements, which would generally be both slower than the current approach and less capable of handling variations in the micro-scale geometry.

Weaknesses: There is not a lot of wiggle room in the paper, but if there were a few sentences room, it would be helpful to put this in the context of other work on computational homogenization methods / multi-scale finite element methods. There is also an important limitation (not directly acknowledged here) associated with micro-scale buckling, as occurs in some of the examples. Depending on the loading path to reach the final geometry of one of the macro-elements, one might have pieces of the micro-structure buckle in different directions, and hence reach different equilibria with different energies -- even though the deformed boundary geometry is the same. Hence, the coarse-grain behavior might exhibit hysteretic effects even though the fine-grain model does not. The usual way of dealing with this is to add some type of history variables (if it is a significant effect) or to ignore it (if the effect is not too significant). I do not think this is a critical weakness, but it would be nice if it were addressed in some way in the paper.

Correctness: Yes.

Clarity: Yes.

Relation to Prior Work: Yes.

Reproducibility: Yes

Additional Feedback: --- Added after rebuttal --- I remain strongly in favor of this paper after the rebuttal.


Review 2

Summary and Contributions: This work presents a method to approximate solutions to PDEs relevant to finding configurations which minimize stored energy when modeling mechanical meta-materials. The proposed method, Composable Energy Surrogates (CES), collapses the problem into a set of spatially local components and modeling the energy in a component given boundary conditions with a neural network surrogate. The surrogates are trained to predict the solutions to PDEs in a small spatial region given shape parameters and boundary conditions. The surrogates are then composed to approximate the energy of larger cellular meta-materials, which can then be optimized over the system configuration. This method is compared with Finite Element Analysis (FEA), and is found to improve efficiency for similar FEA accuracies and improve accuracy over FEA for a given computation budget.

Strengths: This paper introduces an interesting approach to breaking down a complex energy minimization problem into a set of smaller, and composable, components and using neural networks as surrogates to provide an amortized solutions to the smaller component. By modeling boundaries with splines, and utilizing gradient and Hessian information in the surrogate training procedure, the surrogates can be used effectively in the outer optimization problem, and can help avoid running large scale FEA. Thus this work nicely breaks down the large PDE solving problem into a setting which allows for amortization and improving efficiency, and presents a scheme for utilizing neural networks to aid in this task. This work thus shows the use ML and neural network models to approach solving a specific, but relevant, class of PDEs in a new way. The empirical evaluation of the method is limited, but presents strong results. The direct discussion of limitations and opportunities is very helpful to the reader, and strengthens the case that there are interesting directions of future work.

Weaknesses: The empirical evaluation would benefit from additional exploration. For instance, the outer optimization may be sensitive to the quality of the surrogate model energy predictions and gradients. There is little presentation on the quality of the surrogate model predictions and gradients. Is it understood how sensitive the outer optimization is to the accuracy of the surrogate gradients? Even if the gradients are biased, can the outer optimization still find reasonable solutions? In terms of generalization, do you know how the CES method scales with system size in terms of accuracy and evaluation time? If the surrogate models have imperfect energy and gradient predictions, does the use of a composable approach exaggerate such errors? Since the same surrogate is used for each cell, are there correlated biases that may make scaling the system to larger sizes hard? Can the method generalize in terms of shapes and strains / compressions / tensions, in the sense that if you train on a range of these values, can CES find accurate solutions outside of the training range? The structure of the neural network and the optimization procedure for the surrogate loss are not discussed clearly in the paper. It is mentioned that the relevant hyper parameters need further study, but some discussion of the methods sensitivity to the network model and the optimization procedure would benefit the reader. While the use of surrogates to approximate solutions to the inner optimization problem is discussed, the method to solve the outer optimization over boundary conditions to find minimum energy solutions to the composed surrogates is not clearly discussed. Equation (7) give a broad picture, but the details of how this is done with surrogates is not clear. There is mention of using L-BFGS in the evaluation section, and mention of a challenge in this optimization which gives rise to the need for using DAGGER. However, the problem giving rise to the need for the DAGGER approach is not clear. In addition, its not clear how the final optimization problem couples the surrogates. Are the boundaries conditions shared when a boundary is common between two cells? Are any constraints or regularizations used in this optimization? Is the boundary constraint in equation (7) only defined on the macroscopic system boundary, or also on all subregion boundaries? It seems from equation (7) that the constraint is only on the boundary of the full system, so how is that translated into a constraint for subregions? I believe that more clarity is needed for the reader on this topic. Its not clear how the HMC and PDE solver are used together. Given a PDE solution, when you run HMC using such a solution, are you guaranteed to find another minimum energy solution for a different boundary condition and shape parameters? Do I understand correctly that the HMC samples are used as initial conditions which are used in a different run of PDE solver with different boundary conditions and shape parameters? It would benefit the text to be more clear here. In addition, does the HMC require a significant burn-in time before producing reasonable samples? The discuss of the surrogate and i.i.d. assumptions (starting line 266) is not clear, and as a result it is not clear what the role of DAGGER is. There is an explanation in terms of analogies with agents and actions, but the analogy is not clear. Rather than discussing this topic through analogy, this section would benefit from a more concrete disruption of the issue at hand within the context of CES. This may be aided by adding a more clear discussion of the outer optimization and pointing out where the problem arises. ------------------------- After Author Response: Dear Authors, thank you for your responses, I am happy with your responses. I maintain that this is a good submission and vote for accepting

Correctness: The empirical methodology appears correct from the presented material. More discussion on certain aspects of the model are needed.

Clarity: The clarity of the paper needs improvement. Several aspects of the method are not clearly discussed. As previously mentioned, this include the description of how the outer optimization is solved, what neural networks and optimization procedures are used for the surrogate, how the HMC is used in concert with the PDE solver, and a description of the challenges requiring the DAGGER method and how DAGGER is used within the context of CES.

Relation to Prior Work: The difference between this work and other related work is discussed

Reproducibility: No

Additional Feedback:


Review 3

Summary and Contributions: The paper proposes a method to learn the elastic energy minimizer for a cell with given geometry. A surrogate energy model is constructed by learning the minimum energy and its derivatives with respect to the boundary condition. The model is then applied to compute the energy minimizer on a larger structure that consists of repeated cells. The proposed method accelerates the solving of the energy minimization problem compared with the standard Newton-based FEM solver.

Strengths: The method makes use of the repeated structure of metamaterials and reduces the solution to a lower-dimensional problem that can be solved with the help of individual energy surrogate for each cell. Machine learning is an effective way to learn the energy surrogate.

Weaknesses: The comparison uses Newton-based FEM as the baseline, which is known to be slow.

Correctness: There is rigorous derivation of the formulation.

Clarity: The process of using the surrogates to solve the original problem can be explained in more details. In particular, the original problem is to find the energy minimizer subjet to the boundary condition for the whole patch. When using the surrogate model, the optimization variables are the displacements on the union B of cell boundaries. Only part of B is the patch boundary. Currently the paper mentions that L-BFGS is used to solve the original problem via the surrogate models. Does it mean that the variables corresponding to the path boundary are fixed, and the remaining variables are optimized? Please clarify.

Relation to Prior Work: Yes.

Reproducibility: Yes

Additional Feedback: The comparison with the baseline method looks promising. On the other hand, Newton method is neither the fast nor the most stable. As mentioned by the author(s), the baseline does not represent state of the art. Given that the intention of this method is to speed up the calculation, a comparison with more sophisticated methods would be helpful. For example, in the field of computer graphics, there is a particular demand for fast simulation of elastic materials. The method in the following paper could be used to solve the problem in this paper on a discretized domain: - Liu, T., Bouaziz, S. and Kavan, L., 2017. Quasi-newton methods for real-time simulation of hyperelastic materials. ACM Transactions on Graphics (TOG), 36(3), pp.1-16. The method can be adapted to the current problem by ignoring the momentum energy term in their target function. The results in their paper show that it can solve energy minimization problems with up to 100K variables in almost real-time. I wonder how its performance compares with the proposed method here. Update: the authors' response has addressed my questions and I have upgraded my score to accept.


Review 4

Summary and Contributions: This paper presents an efficient way to simulate structured materials with deformable cells by decomposing the total energy as a sum of energies of individual cells. They replace the original energy optimization by a determination of boundary shapes for all cells, and learning a function to approximate subcell energies. For given energy error, the method looks to be O(50) x faster than a large FEM solution. L2 errors in spline approximations to subcell shape are either faster or more accurate than global FEM solutions of similar number of variables or compute time. The paper investigates using neural nets to speed up potential energy surface approximations for a case where exact decomposition holds, and compares with accurate baseline calculations.

Strengths: I liked the presentation of the mathematics in terms of a general method involving collapsed objectives to avoid optimizing in an orginal, larger, basis. The presentation and problem formulation was quite clear, and the relation to existing techniques was adequate. The work is situated within the broader class of applying neural nets to approximate potential energy functions. The example they use is an interesting case where the energy decomposition and independence assumptions should hold very well. I found the use of gradient- and curvature-based losses (justified by ablation studies in the appendix) to be an interesting feature.

Weaknesses: A number of weaknesses are straightforwardly presented in Sec. 8. The appendix addresses many of the main weaknesses of the paper and choice of appendix material was good. There is one outlier in L2 compression that was quite bad (more clear in the appendix) with no comment about this outlier. Is there something about this case that made it particularly difficult, given that the other solutions are visually very convincing? Neural networks are not the focus of the paper; however, the paper remains relevant to NIPS by showing a case where embedding a neural network into a larger framework of decomposable energies and auxiliary variables make good sense.

Correctness: I had no issues with the methodology. The application seems one where Eq. (3) should hold exactly, which made the paper more interesting to me.

Clarity: The paper is well written.

Relation to Prior Work: Relation to previous contributions is well described. The chosen application is nice because many of the underlying assumptions actually should hold. Similar hierarchical energy decompositions are often used even when Eq. 3 only approximately (or even rather poorly!) holds, so in this respect this work is an interesting fundamental experiment showing the abilites of learning a decomposable energy function that introduces an auxiliary basis. A short comment might help the reader better situate the present work within the more usual (much less idyllic) context of approximating potential energies. I was satisfied with the author rebuttal. I still feel it is a good paper and should be published.

Reproducibility: Yes

Additional Feedback: For the FEA example application, I idly wondered what the dimensions of y and x subspaces were, and how many neural net parameters were used to approximate the x energy. All required info is present, but I was lazy. Although the methods seems fully disclosed, there are enough tricky code pieces that it would be non-trivial for me to reproduce their results. line 304 : solves --> solvers

[Author Response · NeurIPS 2020]

Thank you for your valuable feedback, which is very helpful in improving the paper. We're encouraged by the broadly positive feedback, and greatly appreciate the critical and constructive suggestions.

**Reviewer 1:** *"Put this in the context of other work on computational homogenization / multi-scale finite element methods"* We will squeeze this into the paper. Our method is related to these and the boundary element method (BEM). It can be seen as a way to learn a neural BEM for problems with no analytic BEM, or a way to do multi-scale FEM / direct macro-micro homogenization (Schroeder 2014) but replacing the fine-resolution FEM with a NN and doing the fine-to-coarse averaging somewhat differently.

*"Limitation associated with micro-scale buckling... the coarse-grain behavior might exhibit hysteretic effects":* Good observation: our model will not capture hysteretic effects in the "true" coarse-grained behavior which occur due to "latent" micro-scale displacements. No problem if you just want to find an energy minimizer, but a problem if you want to understand particular loading/solution paths. We will mention this.

**Reviewer 2:** *"How sensitive is the outer optimization to the accuracy of the surrogate gradients?"* Qualitatively, the surrogate gradients' accuracy is very important - the outer optimization produced poor results without Sobolev training and DAGGER. Quantitatively, measuring the sensitivity of the outer optimization to accuracy of the surrogate, and how to design surrogates which are more "robust" to outer simulation, are important questions for future work.

*"Do you know how the CES method scales with system size in terms of accuracy and evaluation time":* In terms of evaluation time, CES scale linearly with the number of cells in the composed solid. In terms of accuracy of the obtained solution, this will depend on the particular macro task, and measuring this scaling we leave to future work.

*"the method to solve the outer optimization over BCs to find minimum energy solutions to the composed surrogates is not clearly discussed":* Thanks for the feedback - we will try clarify this. We impose Dirichlet BCs by fixing the corresponding DoFs (DoFs = blue points in Fig 2). The energy in each cell is computed using the DoFs on the cell boundaries. Free DoFs are optimized to minimize total predicted energy using LBFGS.

*"The discuss of the surrogate and i.i.d. assumptions (starting line 266) is not clear, and as a result it is not clear what the role of DAGGER is":* Thanks for the feedback - we will try clarify this. The surrogate will only be accurate on cell BCs similar to training data. Given an arbitrary training set, the cell BCs encountered while solving the composed solid might be dissimilar to the training set. We use DAGGER to adjust the training set to match states encountered while solving, for a distribution over "reasonable" macro-problems.

*"Are the BCs shared when a boundary is common between two cells":* Yes. We have 1 DoF for each blue point in Fig 2.

*"Its not clear how the HMC and PDE solver are used together":* HMC is used to generate training BCs, preferring larger strains with lower energies (to explore pore-collapse behavior, avoiding non-physical / exploding-energy displacements). The PDE solver is used to compute the gradient of the pdf (which depends on E) w.r.t. the BC. This grad is needed each leapfrog integration step. Given BCs, we run the solver to determine the internal u and E. We compute dE/dBC with the adjoint method. Then we use this to compute the gradient of the pdf w.r.t. the BCs, needed for the leapfrog step.

*"does the HMC require a significant burn-in time before producing reasonable samples":* No. Note: we don't truly care about drawing samples from any given distribution: we just want to obtain a variety of data which explores buckling modes while avoiding non-physical BCs. This initial data is not that important as we use DAGGER to adapt the dataset to the task at hand. Per appendix, HMC took between 3 and 100 leapfrog steps per sample. We save the resulting BC regardless of the outcome of the accept/reject step (which determines the starting point for the next leapfrog integration).

**Reviewer 4:** *The process of using the surrogates to solve the original problem can be explained in more detail. Are the variables corresponding to the path boundary fixed, and the remaining variables optimized?* Yes. The variables corresponding to the BC are fixed, corresponding to a Dirichlet BC on the composed solid. The remaining variables are optimized to minimize the (surrogate-predicted) energy.

*Newton method is neither the fast nor the most stable... a comparison with more sophisticated methods would be helpful:* Thanks, we weren't aware of the Liu et al paper. From a brief look it looks like Liu et al's method is tailored for time-stepping problems where you want an explicit result at many densely/evenly-spaced frames. This is very different to statics (or timestepping with adaptive or larger step-sizes) where it may have less advantage vs Newton's. When we initially / informally tried SNES' quasi-Newton method, it was slower solving the static PDE than Newton's method. However we will make sure to read Liu et al carefully and to discuss and benchmark vs both quasi-Newton methods and dynamic relaxation with kinetic damping in the revision.

**Reviewer 5**: *"There is one outlier in L2 compression that was quite bad":* We will discuss this in the main paper.

*"A comment might help the reader situate this work within the more usual (less idyllic) context of approximating potential energies.":* This is a good suggestion: we will relate to other work in learning energies.

*"I wondered what the dimensions of subspaces were, and how many NN params were used":* The MLP had 3 layers of 128 units. The reduced space was 72d for a 2x2 cell and 690d for a 8x8 composite. Data was gathered with FEM of around 5000d for 2x2 cells (dep. on pore shape). Sizes of each composite FEM baseline are in the appendix.

*"There are enough tricky code pieces that it would be non-trivial for me to reproduce their results":* Yes, there are quite a few moving parts here (largely due to using both Fenics and Pytorch). We provide our code to help with reproducibility.

[Meta-Review · NeurIPS 2020]

The paper presents a very nice application of ML techniques to a new problem domain and has the potential to open a new research direction. The authors show how to use neural nets to solve a specific class of PDEs in a novel way. Their technique is elegant and more efficient than traditional finite element analysis. The work is well grounded both in ML and the application domain of computational mechanics. The main issues raised by the reviewers concern clarity and they have been addressed in the authors' rebuttal.